**Data Availability Statement:** Data cannot be shared without restrictions because they are

# Changes to the sample design and weighting methods of a public health surveillance system to also include persons not receiving HIV medical care

Christopher H. Johnson[1], Linda Beer[1]*, R. Lee Harding[2], Ronaldo Iachan[2], Davia Moyse[2], Adam Lee[2], Tonja Kyle[2], Pranesh P. Chowdhury[1], R. Luke Shouse[1]

**1** Division of HIV/AIDS Prevention, Centers for Disease Control and Prevention, Atlanta, Georgia, United States of America, **2** ICF International, Inc., Calverton, Maryland, United States of America

* LBeer@cdc.gov

## Abstract

### Objectives

The Medical Monitoring Project (MMP) is a public health surveillance system that provides representative estimates of the experiences and behaviors of adults with diagnosed HIV in the United States. In 2015, the sample design and frame of MMP changed from a system that only included HIV patients to one that captures the experiences of persons receiving and not receiving HIV care. We describe methods investigated for calculating survey weights, the approach chosen, and the benefits of using a dynamic surveillance registry as a sampling frame.

### Methods

MMP samples adults with diagnosed HIV from the National HIV Surveillance System, the HIV case surveillance registry for the United States. In the methodological study presented in this manuscript, we compared methods that account for sample design and nonresponse, including weighting class adjustment vs. propensity weighting and a single-stage nonresponse adjustment vs. sequential adjustments for noncontact and nonresponse. We investigated how best to adjust for non-coverage using surveillance data to post-stratify estimates.

### Results

After assessing these methods, we chose as our preferred procedure weighting class adjustments and a single-stage nonresponse adjustment. Classes were constructed using variables associated with respondents' characteristics and important survey outcomes, chief among them laboratory results available from surveillance that served as a proxy for medical care.

### Conclusions

MMPs weighting procedures reduced sample bias by leveraging auxiliary information on medical care available from the surveillance registry sampling frame. Expanding MMPs

collected under a federal Assurance of Confidentiality. Data are available from the US Centers for Disease Control and Prevention for researchers who meet the criteria for access to confidential data. Data requests may be made to the Clinical Outcomes Team in the Division of HIV/AIDS Prevention at the Centers for Disease Control and Prevention, 1-404-639-6475.

**Funding:** Funding for the Medical Monitoring Project is provided by a cooperative agreement (PS15-1503) from the US Centers for Disease Control and Prevention (CDC). CDC has a contract with ICF International, Inc. for operational and technical support to conduct the Medical Monitoring Project The funder provided support in the form of salaries for all authors, but did not have any additional role in the study design, data collection and analysis, decision to publish, or preparation of the manuscript. The specific roles of the authors are articulated in the 'author contributions' section.

**Competing interests:** Funding for the Medical Monitoring Project is provided by a cooperative agreement (PS15-1503) from the US Centers for Disease Control and Prevention (CDC).CDC has a contract with ICF International, Inc. for operational and technical support to conduct the Medical Monitoring Project. This affiliation does not alter our adherence to PLOS ONE policies on sharing data and materials.

population of focus provides important information on characteristics of persons with diagnosed HIV that complement the information provided by the surveillance registry. MMP methods can be applied to other disease registries or population-monitoring systems when more detailed information is needed for a population, with the detailed information obtained efficiently from a representative sample of the population covered by the registry.

## Introduction

The Medical Monitoring Project (MMP) is a Centers for Disease Control and Prevention (CDC) surveillance system that provides population-based information on behaviors and clinical characteristics of persons with diagnosed HIV [1]. The information collected through MMP includes essential information for preventing HIV-related morbidity and HIV transmission, such as barriers to medical care utilization, adherence to treatment, and sexual behaviors [2].

During 2005–2014, MMP sampled persons from HIV care facilities, which excluded persons who had not been linked to or retained in HIV care [3]. Persons with undiagnosed HIV or diagnosed but not receiving medical care were estimated to account for 92% of HIV transmissions in 2009 [4], and ensuring they receive medical care is a national HIV prevention goal. In 2015, MMP expanded its population of inference to all adults with diagnosed HIV, using CDC's National HIV/AIDS Surveillance System (NHSS) to construct a frame representing this population. NHSS is an HIV case surveillance registry first established in 1981 that collects a core set of information on the characteristics of all persons with diagnosed HIV in all U.S. states and dependent areas [5].

### 2005–2014 population, frame, and sample design

MMP was designed to produce nationally representative estimates as well as locally representative estimates for participating project areas [6]. States were the primary sampling units (PSUs) and were sampled with probability proportional to size, with the number of AIDS cases reported through the end of 2002 used as the measure of size, resulting in some states' being selected with certainty. Within sampled states, six jurisdictions with federally funded local HIV surveillance programs brought the total number of independent project areas, or strata in the national sample design, to 23.

The 2005–2014 MMP population of inference implicitly excluded those not receiving care because NHSS was not comprehensive in all states (e.g., before HIV without an accompanying AIDS diagnosis was reportable in every state) and was thus inadequate as a sampling frame. Instead, MMP employed multi-stage probability-proportional-to-size facility-based sampling and generated patient lists for participating facilities, from which patient samples with probability inversely proportional to reported patient volume were drawn. Despite the advantages of constructing a frame in stages, this sampling method was labor- and time-intensive and excluded a key population, persons not receiving HIV care [7].

### 2015—Present population, frame, and sample design

Each state and territory in the US collects name-based HIV and AIDS case surveillance registry data that include HIV-related laboratory tests, which provide information on HIV care utilization and disease progression. These data are reported to NHSS, after which CDC cleans and de-duplicates records [8]. Completeness and timeliness of reporting having improved in the

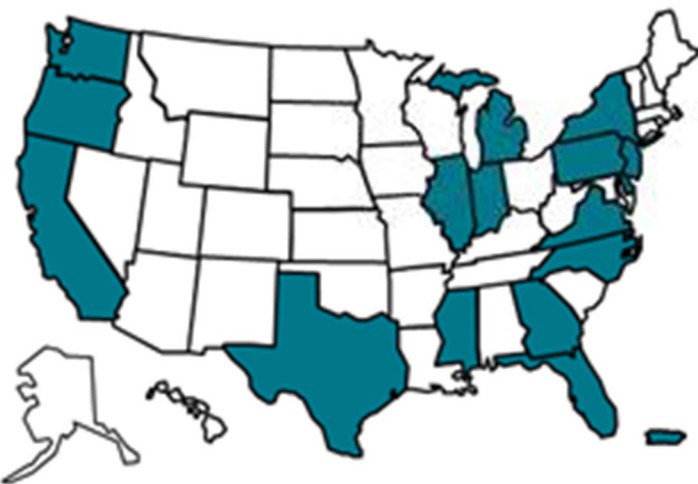

**Fig 1. Map of areas participating in MMP.** Note: These 23 areas were funded to conduct data colloection for the 2015 cycle: California (including the separately funded jurisdictions of Los Angeles Country and San Francisco), Delaware, Florida, Georgia, Illinois (including the separately funded jurisdiction of Chicago), Indiana, Michigan, Mississippi, New Jersey, New York (including the separately funded jurisdiction of New York City), North Carolina, Oregon, Pennsylvania (including the separately funded jurisdiction of Philadelphia), Puerto Rico, Texas (including the separately funded jurisdiction of Houston), Virginia, and Washington.

decade following the initiation of MMP, and NHSS comprises the most comprehensive source of information for the population of interest. The NHSS surveillance registry data submitted to CDC as of December 31, 2014, served as the sampling frame for the 2015 MMP cycle (the first data collection cycle under the new design).

Despite fundamental changes to sampling at the local level, MMP retained the same sample of states selected in 2005 in the new sample design (Fig 1). An analysis of counts of reported 2011 HIV diagnoses showed that the proportional contribution of states to the burden of HIV had not changed appreciably from the distribution of AIDS cases in 2002. Thus, the first-stage design weights reflecting states' original sampling probabilities were still reasonably close to what they would have been if sampled using more recent HIV diagnosis data and were retained [9]. When weighting probability samples, some practitioners prefer to adjust respondents' data to comport with population totals via a calibration process [10], rather than through separate adjustments for selection probability and post-stratification. Absent compelling reasons to change the stages of adjustment, we chose to keep weighting methods the same as previously employed (e.g., post-stratification as a separate adjustment).

In 2015, the sample design and frame of MMP changed from a system that included only HIV patients to one that captures the experiences of persons receiving and not receiving HIV care. In this manuscript, we describe methods investigated for calculating survey weights for the 2015 cycle, the approach chosen, and the benefits of using a dynamic surveillance registry as a sampling frame. Such a comprehensive description of the methods, along with various options considered and ruled out, as well as the rationale for these decisions, has not previously been available. This information can inform other studies that use sample survey methods.

The 2015 to present national population of inference for MMP is all adults with diagnosed HIV living in the US, and eligibility criteria reflect these criteria (i.e., alive, diagnosed with HIV, aged 18 years or older, residing in the US). Information on presumed current residence was used to construct separate frames from NHSS data for each of the 23 participating project areas.

Records in NHSS are de-identified (under provisions of CDC's Assurance of Confidentiality [11]) and include only limited information about where the person currently resides. CDC staff drew simple random samples from the 23 separate frame files, and project area staff linked their samples to local case surveillance systems and extracted more-detailed contact information for locating and recruiting sampled persons. Sample sizes deemed sufficient under the old design ranged from 200 to 800, and precision was expected to increase with unclustered samples (in compared with the previous multi-stage, clustered sample design).

## Materials and methods

In this manuscript, we present a methodological study of MMP weighting methods. Weighting respondents' data incorporates three adjustments, correcting for different sampling probabilities, nonresponse, and frame limitations (Table 1). The successive adjustment factors are multiplied together to derive analysis weights. A novel feature of MMP is the creation and use of different sets of weights for national and local estimates [12].

### Design weighting

The first component of the weight is the design weight, the reciprocal of the probability of selection. This component is important for national weights but not applied to weights for

**Table 1. Components of MMP analysis weights.**

| COMPONENT | FORMULA | COMMENT |
|---|---|---|
| **DESIGN WEIGHTING STAGE** | | |
| Design weight | $\hat{W}_{0j} = \begin{cases} \dfrac{1}{P_j} & \text{Project areas} \\ \dfrac{1}{P_i P_j} & \text{National} \end{cases}$ | For individual $j$ sampled with probability $P_j$. National weights additionally incorporate first-stage sampling of state $i$ with probability $P_i$. |
| Multiplicity adjustment | $\hat{W}_{1j} = \begin{cases} \dfrac{\hat{W}_{0j}}{2} & \text{if sampled more than once} \\ \hat{W}_{0j} & \text{if sampled only once} \end{cases}$ | |
| **NONRESPONSE ADJUSTMENT STAGE** | | |
| Overall nonresponse weight adjustment factor | $W_{2jk} = \dfrac{\sum_{j \in A} \hat{W}_{1jk}}{\sum_{j \in R} \hat{W}_{1jk}}$ | Adjustments made within nonresponse classes $k$, which vary by area as well as nationally. A is all eligible sampled persons; R is respondents only. |
| Overall nonresponse adjusted weight | $\hat{W}_{2j} = \hat{W}_{1j} W_{2j}$ | |
| **POST-STRATIFICATION AND TRIMMING STAGE** | | |
| Initial post-stratification factor | $W_{3jh} = \dfrac{T_h}{\sum_{j \in R} \hat{W}_{2jh}}$ | Adjustments made within cells $h$ defined by gender, race/ethnicity, and age, by area and nationally. $T$ is the number of eligible persons on the delayed frame; $R$ is respondents only. |
| Post-stratified weight | $\hat{W}_{3j} = \hat{W}_{2j} W_{3j}$ | |
| Trimmed weight | $\hat{W}_{4j} = \begin{cases} \text{Median } \hat{W}_{3j} + 4(\text{IQR}) & \text{if } \hat{W}_{3j} > \text{Median } \hat{W}_{3j} + 4(\text{IQR}) \\ \hat{W}_{3j} & \text{if } \hat{W}_{3j} \leq \text{Median } \hat{W}_{3j} + 4(\text{IQR}) \end{cases}$ | |
| Final post-stratification factor | $W_{5jh} = \dfrac{T_h}{\sum_{j \in R} \hat{W}_{4jh}}$ | Within cells $h$ defined by sex, race/ethnicity, and age, by area and nationally. |
| **FINAL WEIGHT STAGE** | | |
| | $\hat{W}_{5j} = \hat{W}_{4j} W_{5j}$ | |

project areas, for which analysis is conditional on their initial selection and the implied value of the factor is 1. Selection probabilities were uniform within jurisdictions but varied greatly across states—areas with fewer cases had higher sampling rates, and conversely.

The data reconciliation process disclosed some duplicate records on the frame, where multiple records were found to represent the same person. If a duplicate record was sampled, its weight was reduced by half to reflect its multiple opportunities for selection; we capped this multiplicity adjustment at 2 because finding more than one duplicate record was rare. Duplicate records not sampled did not require adjustment, although their contributions to frame totals were correspondingly reduced. Through interview or NHSS updates, some sampled persons were found to have lived in a different project area at the time of sampling. Their records were reassigned to the other project area, but retained the original design weight.

### Nonresponse adjustment

The second component of the weight is nonresponse adjustment, which is analogous to the reciprocal of the probability of responding. An advantage of the new design is the availability of extensive demographic data from NHSS at the time of frame construction. Using variables significantly associated with nonresponse in bivariate analysis, we conducted multivariable analysis to identify predictors of nonresponse at national and project area levels. We considered these predictors of nonresponse: sex at birth, age, race/ethnicity, residency (US vs. other, and MMP vs. non-MMP jurisdiction), transmission risk category, AIDS at HIV diagnosis, time since last update of contact information, time since diagnosis, most recent viral load measurement, and presumed HIV care status (a three-level measure based on HIV lab results in NHSS: 2+ HIV labs in the past 12 months 90 days or more apart, at least 1 HIV lab in the past 12 months, and 0 HIV labs in the past 12 months). When an adjustment method required us to choose the strongest predictors from those that remained significant in multivariable analyses, we ranked them by absolute log odds ratio ($|\log(OR)|$, the absolute value of the beta estimate), based on parameter estimates from the final models.

### Adjustment method

Previously, MMP weighting employed the weighting class method, in which a few adjustment classes were formed based on variables found in logistic regression analysis to predict nonresponse. The change in MMP sampling methods was an opportunity to investigate other methodological changes, such as whether another weighting method previously explored in national surveys might perform better [13, 14]. Based on weighting methods used in a pilot project of MMP's new design [15], MMP considered adopting the propensity weighting method. This method allows incorporating more predictors than the weighting class method (for which we could use only the strongest predictors, at most two per project area, lest the classes formed by the resulting cross-classification become too sparse); propensity weighting is a model-based generalization of the weighting class method that permits an arbitrary number of predictors, including continuous variables [16]. The resulting predicted probabilities are then typically grouped into a few categories, often quintiles, to reduce their variability. We implemented both methods to calculate weights, and compared the resulting estimates and their variance.

### Adjustment stages

For each adjustment method, we investigated making sequential nonresponse adjustments for two outcomes: noncontact and, among those contacted, nonresponse. Even though contacting respondents is necessary before they can be interviewed, most surveys collapse these stages

rather than applying differential adjustments that are multiplied together. Our hypothesis was that different factors might be associated with contact compared with response among those contacted, and that by making separate adjustments in sequence (implying two adjustment factors multiplied together in a respondent's weight) we might reduce total nonresponse bias.

## Noncoverage adjustment

The study population for the MMP 2015 cycle was all persons with diagnosed HIV at the end of the reference year (i.e. December 31, 2014). A year after the MMP data collection cycle ended, we constructed a second, "delayed" frame that included records that would have been eligible if they had been reported at the time of sampling. The dynamic nature of the frame also allowed us to identify cases determined to be ineligible after the sampling date and adjust the population size accordingly. This updated information on the population of persons with diagnosed HIV was used for post-stratification to known totals, correcting for frame limitations.

Population characteristics typically used for post-stratification noncoverage adjustment are those that correlate with key outcome measurements, but are unavailable for nonresponse adjustment. However, in MMP demographic and care-related variables were available for the entire sample, regardless of response. A count of delayed frame records provided updated population size estimates by sex, age, and race/ethnicity. Post-stratifying to these totals forces the sample-based estimate of population size to conform while correcting for late reports and updated eligibility information. The final post-stratification adjustment provided additional protection against the possibility that the sample-based nonresponse adjustments had distorted demographic distributions.

Post-stratification also incorporated a weight-trimming process that limited the weights' variability and thereby the variance of estimates. Within adjustment cells, initial weights were compared to the median weight plus 4 times the interquartile range and truncated if they exceeded this cap. These capped weights were then post-stratified to re-adjust and increase any weight sums reduced by trimming. However, due to the intensive follow-up necessary for MMP recruitment in addition to the continual post-sampling updating of the surveillance registry data, enhanced information revealed ineligible records and inaccurate residence classification, and adjusting for these reduced the estimated population totals.

## Variance estimation

Calculating survey estimates requires application of appropriate weights and, for standard errors to accompany point estimates, application of appropriate design variables. We developed strata and cluster variables that accounted for the sample design. Because of the two-stage, stratified sample design, different sets of sample design variables are employed for variance estimation at the national and local levels (an unusual feature of MMP).

Nationally, many states (which were the PSUs in the stratified probability proportional to size design) were sampled with certainty because of their large numbers of persons living with AIDS, and each of these was defined as its own stratum. Among non-certainty PSUs, strata were created by grouping 2–3 states that had similar selection probabilities. To provide stratum-level between-cluster variance components, all strata needed at least two clusters. For certainty PSUs, patients were the clusters. For the strata composed of non-certainty states, the state was the cluster.

For project area estimates, variance estimation is conditional on the initial sampling of states as PSUs, meaning that this stage of sampling is ignored and the design is adequately described as a simple random sample. The sampling fraction exceeded 10% in only the smallest

state sampled, Delaware, and we determined there was no need to apply a finite population correction factor [17] in any area.

In accordance with guidelines for defining public health research, CDC and most project areas have determined MMP is public health surveillance used for disease control, program, or policy purposes. Local institutional review board (IRB) approval is obtained from the University of Puerto Rico Medical Science Campus IRB and the Virginia Department of Health IRB. Written or documented informed consent is obtained from all participants, as required by local areas.

## Results

### Respondents

Weighted characteristics of adults with diagnosed HIV from the 2015 MMP cycle have previously been reported [2]. In brief, an estimated 75% (95% confidence interval [CI]: 72.1–77.4) were male, 48% (CI: 44.1–51.3) identified at heterosexual or straight, and 41% (CI: 31.0–51.4) were Black or African American. An estimated 62% (CI: 58.8–64.9) had received an HIV diagnosis at least 10 years earlier.

### Response rates

Of 9,700 persons sampled from an initial frame count of 782,718, 521 were determined to be ineligible (5.4%; project area range 1.5%–15.9%, Fig 2). Of these, 299 died before the sampling date (but their death had not yet been reported to NHSS), 356 lived outside an MMP jurisdiction, 40 had no HIV diagnosis, 4 were duplicates of another sampled person, and 1 was less than 18 years of age.

The national response rate was 39.8% (range 30.8%–48.7%). Of the initial eligible sample of 9,179, there were 5,525 eligible nonrespondents (60.2%). Of these, 1,459 (15.9%) were contacted but either refused or did not respond to contact attempts, while 4,066 could not be located and were never contacted (44.3%). The national contact rate among eligibles was 55.7% (range 38.8%–79.8%). Of the 5,113 who were both eligible and contacted, 3,654 responded; thus cooperation (i.e., response among those contacted) was 71.5% nationally (range 55.7%–91.6%). Response rates in 2015 were generally comparable to those experienced in earlier data collection cycles under the old design [16].

A concern about the new design was that contact information might be outdated or incorrect, particularly for those diagnosed with HIV years earlier. A small proportion of respondents, 3.7%, were found to have lived in a different MMP jurisdiction on the sampling date than the jurisdiction from which they were sampled. Using information from the delayed frame, we found that almost 3.6% of persons were ineligible because they had moved out of the sampling jurisdiction before the sampling date or, more rarely, had never lived in the sampling jurisdiction. Because many sampled persons were never contacted, however, the actual percentage who moved may be higher. Challenges in fielding the MMP sample reflect the limitations of case surveillance data due to ineligibility or insufficient location information.

### Nonresponse adjustment

In our nonresponse analyses, the strongest predictor of response, nationally and in 17 project areas, was presumed HIV care status. Measured by absolute log odds ratios, this effect ranged across areas from 0.472 to 0.978 when comparing those not receiving care to those in care (corresponding to odds ratios range 0.624–0.376). Because a person's care status is dynamic and laboratory reports can be delayed, we used updated information from NHSS to measure care

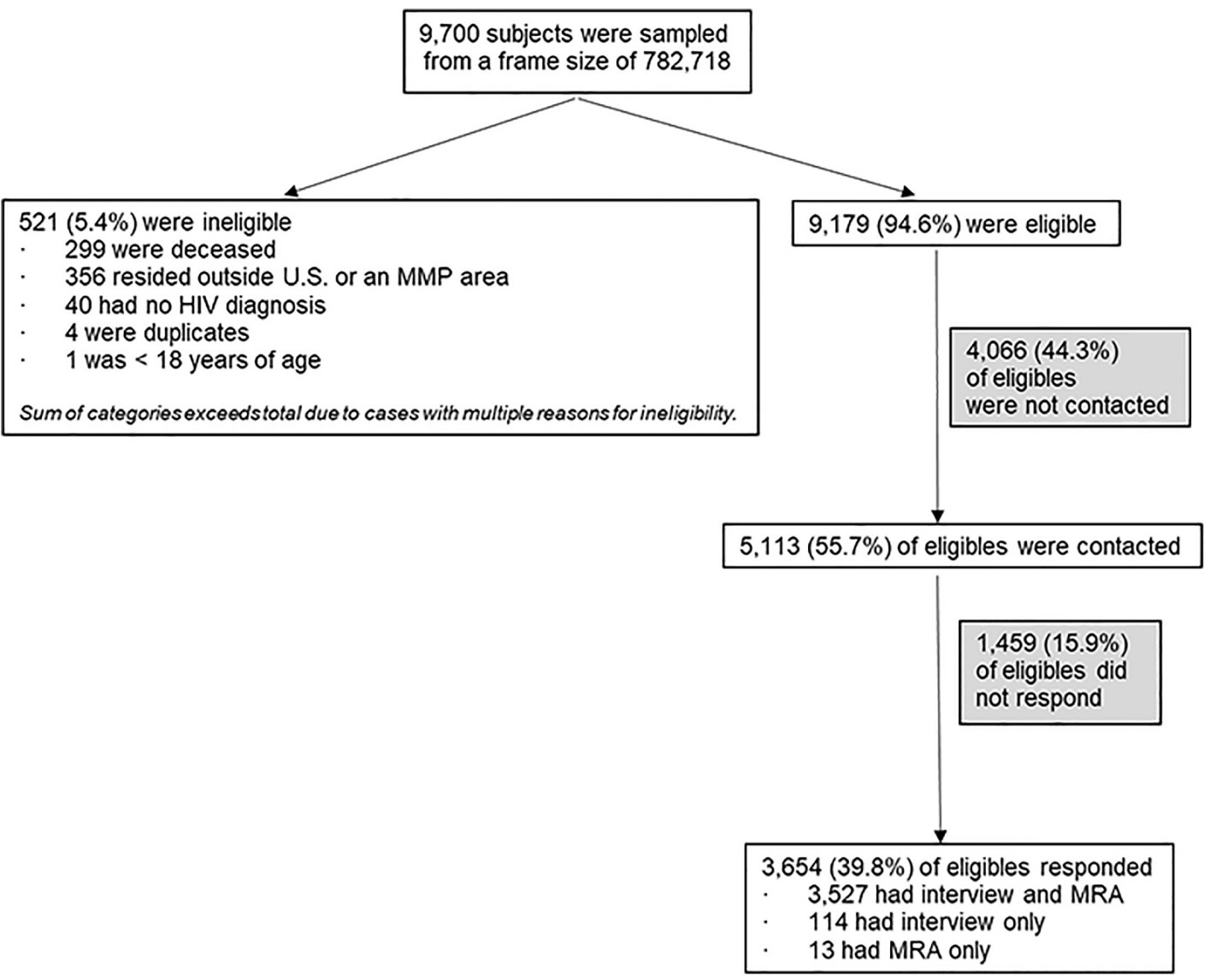

**Fig 2. MMP response rates.**

status in the period preceding a respondent's interview (and, for nonrespondents, their care status in the period before the median time of interview for respondents) rather than at the time of sampling.

The adjustment variables varied across areas in both methods, but the weighting class and propensity models were most often equal, using the same set of categorical adjustment variables. Locally, few continuous variables exhibited strong associations with response, leading to identical models in some cases (and only slightly different models when grouping propensities into quintiles resulted in collapsing categories that were distinct in the corresponding weighting class model for an area). In the final models, care status was significant in a majority of areas, and in 10 cases no additional factor was significant. Although the model fit statistics were comparable for both methods (see S1 Appendix), we chose the cell weighting method because, having less variance, it performed better than the propensity method.

For both models, single-stage and two-stage adjustments produced similar results for national estimates, both for the resulting weighted estimates themselves and the variance added due to weighting. Locally, if one or more variables was significantly associated with non-contact, in many cases (11 of 22 areas) no variable was associated with nonresponse—possibly due in part to the reduced number of persons contacted in areas with low contact rates. In one instance, no variable was associated with noncontact, but variables were associated with nonresponse. Although we did not find this using these data, the same factor could operate in opposite directions across the two stages (e.g. decreasing contact but increasing response), and would thus have an attenuated effect on total response, since the adjustment factors are multiplied together and the final factor reflects their cumulative effect. Because implementing a two-stage adjustment was generally not feasible locally, we opted for a single-stage adjustment nationally.

## Discussion

The 2015 data collection cycle expanded the scope of MMP. As expected, including persons diagnosed with HIV but not receiving care added a harder-to-reach population. Many of the key outcomes monitored by MMP relate to receipt of care. Variables strongly correlated with both key outcomes and response, such as care status, are normally ideal candidates for use in nonresponse adjustment. They must also be available for both respondents and nonrespondents, and are usually measured when the sample is drawn. Their utility for this purpose in MMP, however, is somewhat limited by the dynamic nature of care. Many people classified as out of care when sampled had in fact received care, but lab results associated with visits are subject to reporting delays—that is, these people were misclassified, based on the initial information. Information from the delayed frame updated care status for respondents and nonrespondents alike, but medical records abstracted for respondents may disclose care not yet reflected in surveillance records. Thus, this group of respondents is affected disproportionately, as is the efficiency of this information for reducing nonresponse bias. However, our use of a continually updated national HIV surveillance registry as a frame allowed us to partially correct for this. This approach may be useful for other studies of populations for which an important characteristic is dynamic and associated with both response and other key variables measured by the study. In addition, the use of updated information for weighting allowed us to adjust for noncoverage and eligibility, thereby improving the quality of the data beyond what would be possible if we had used only the initial sampling frame.

In addition to misclassification of care status, response was low among those presumed to be out of care according to NHSS, and many of those who responded were later determined to be in care based on interview and medical record abstraction. The nonresponse adjustment factors for this group were so high in certain areas that their weights were capped during the weight-trimming stage. Weight trimming to reduce variance, while potentially increasing bias, is a standard trade-off in weighting and is accepted practice. Still, capping the weights limited out-of-care respondents' contributions to weighted estimates, making weighted estimates for the entire MMP population more similar to estimates for the in-care subpopulation, since the in-care respondents' weights were not reduced.

An appealing aspect of the propensity method was the opportunity it offered to use a common set of predictors in all areas when constructing national weights, bringing greater uniformity to our methods. For the sake of interpretability of the propensity models at the local level, however, we chose to include only those predictors exhibiting significant bivariate associations with the local outcome. This provided more continuity with weighting methods employed in previous MMP cycles. Like the weighting class method, the propensity method also involved a

screening step for bivariate association, so it involved no less effort and offered no logistical advantage for the current MMP design.

We chose to continue to adjust for overall nonresponse using a single-stage adjustment. The two-stage weighting approach conferred little benefit to local estimates. Ultimately, few areas had distinct adjustment factors for noncontact and contacted nonresponse. Motivated by theoretical considerations that were not so much statistical as behavioral and logistical (i.e. that determinants of contact and response might differ), the data did not support modeling such a sequential process. In a different application or with less sparse data, however, a multi-stage adjustment might improve the representativeness of weighted results. Nationally, although sample sizes were large and there were significant and distinct associations for each stage of nonresponse, we judged the two-stage adjustment not worth the additional analytic effort that building separate models required.

After evaluation based on the comparisons of the weighted sums, design effects, and the comparison of the key weighted estimates, we chose to continue to use the cell weighting method, coupled with a single-stage adjustment. For project areas, the cell weighting method had a smaller weighting-induced design effect than the propensity method. For a majority of project areas, both methods found significant predictors for only a single stage of nonresponse.

## Public health implications

Under the old design, MMP studied a subset of persons with diagnosed HIV receiving care and reached them indirectly through multi-stage sampling. Starting with a frame that includes all persons with diagnosed HIV allows for better integration of MMP and NHSS, and this closer linkage benefits both systems. Many survey frames are static, but NHSS is ongoing surveillance, subject to periodic updates, presenting both opportunities and challenges. Updated information on deaths, new case reports, and residence was key to assigning appropriate weights to sampled persons and was used to evaluate the quality of information available on the sampling date.

MMP changed its design to tap into an existing registry whose completeness and timeliness are well established, sparing the considerable, ongoing effort required to create frames of providers and patients. Using NHSS also means that MMP inherits both its strengths and limitations. NHSS is extensively used for reporting HIV prevalence and trends, providing information on a limited number of key characteristics of persons with diagnosed HIV, but was not designed to be a survey frame. Different considerations influenced NHSS's development, and the inherent difficulty of reporting on recent receipt of care makes it less than optimal for purposes of weighting MMP data, which depends on a participant's date of interview during a relatively long field period.

MMP complements the breadth of NHSS, which includes all persons with diagnosed HIV but has limited information about them, with in-depth information from personal interviews and abstracted medical records. Sample surveys using MMP methods could be feasible for supplemental surveillance in other disease registries and population-monitoring systems whose timeliness and completeness are established. Doing so is often more cost-effective than developing new frames [12]. Moreover, the effort required to locate and contact sampled subjects may disclose problems in the routine operation of the underlying system and lead to its improvement.

## Supporting information

**S1 Appendix. AUC statistics by method and project area.**
(DOCX)

## Acknowledgments

We thank MMP participants, project area staff, and Provider and Community Board members. We also acknowledge the contributions of the Clinical Outcomes Team and the Behavioral and Clinical Surveillance Branch at CDC and the MMP Project Area Group Members.

## Author Contributions

**Conceptualization:** Christopher H. Johnson, Linda Beer, R. Lee Harding, Ronaldo Iachan, Davia Moyse, Adam Lee, Tonja Kyle, Pranesh P. Chowdhury, R. Luke Shouse.

**Formal analysis:** Christopher H. Johnson, R. Lee Harding, Davia Moyse, Adam Lee.

**Methodology:** Christopher H. Johnson, Linda Beer, R. Lee Harding, Ronaldo Iachan, Davia Moyse, Adam Lee, Tonja Kyle, Pranesh P. Chowdhury, R. Luke Shouse.

**Supervision:** R. Luke Shouse.

**Writing – original draft:** Christopher H. Johnson, Linda Beer.

**Writing – review & editing:** Christopher H. Johnson, Linda Beer, R. Lee Harding, Ronaldo Iachan, Davia Moyse, Adam Lee, Tonja Kyle, Pranesh P. Chowdhury, R. Luke Shouse.

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
