## [Decision Letter · Decision Letter 0]

5 Nov 2019

PONE-D-19-17946

Changes to the sample design and weighting methods of a public health surveillance system in order to include persons not receiving medical care

PLOS ONE

Dear Dr. Beer,

Thank you for submitting your manuscript to PLOS ONE. After a long and complex process to identify reviewers that could accept reviewing your article (we invited 20 reviewers that did not accept), based on the reviewer that revised your article we feel that it has merit but does not fully meet PLOS ONE’s publication criteria as it currently stands. Therefore, we invite you to submit a revised version of the manuscript that addresses the points raised during the review process.

Please see attached the comments raised by the reviewer. 

We would appreciate receiving your revised manuscript by Dec 20 2019 11:59PM. To enhance the reproducibility of your results, we recommend that if applicable you deposit your laboratory protocols in protocols.io, where a protocol can be assigned its own identifier (DOI) such that it can be cited independently in the future. For instructions see: http://journals.plos.org/plosone/s/submission-guidelines#loc-laboratory-protocols

We look forward to receiving your revised manuscript.

Kind regards,

Omar Sued, MD

Academic Editor

PLOS ONE

Journal Requirements:

2. Please disclose the author affiliation to ICF International, Inc. in your competing interest statement.

* Thank you for stating the following in the Competing Interests section:

"The authors have declared that no competing interests exist.".

We note that one or more of the authors are employed by a commercial company: 'ICF International, Inc'.

Reviewers' comments:

Reviewer's Responses to Questions

**Comments to the Author**

1. Is the manuscript technically sound, and do the data support the conclusions?

Reviewer #1: No

2. Has the statistical analysis been performed appropriately and rigorously? 

Reviewer #1: Yes

3. Have the authors made all data underlying the findings in their manuscript fully available?

Reviewer #1: No

4. Is the manuscript presented in an intelligible fashion and written in standard English?

Reviewer #1: Yes

5. Review Comments to the Author

Reviewer #1: This research aimed to describe methods used for computing survey weights, the approach chosen, and the benefits of using a dynamic surveillance registry as a sampling frame for adults living with HIV from the National HIV Surveillance System in the United States. While the authors have reported an excellent research topic, I believe that the following comments can be also helpful to consider in their revised version. What I believe this manuscript lacks are bare explanations of the methods used in the Methods section and bare presentation of the Results section. The authors should provide a detailed description and presentation of their findings.

Title:

I believe the title should show the area of this research. This is why I think we should have an “HIV” word somewhere like, “… in order to include persons not receiving HIV medical care”. As well, the title can be shorter. For example, “in order to include persons not receiving” can be written as “to also include persons not receiving”. The word “also” (or any other alternative) in this suggested title can reflect that the system had not included such individuals, but these should be also included along with those who receive HIV care.

Abstract:

When the authors claim that “weighting class adjustments and a single-stage nonresponse adjustment performed best,” they have to be more specific in what regard these methods turn to be the “best.” Or, “strongly associated with” does not reflect anything unless we see some outputs or even some key findings. Please avoid using such vague sentences and provide more evidence for these results.

The abstract does not seem to be a good place talking about how the results of this HIV case study might also have implications for “other disease registries”. This can be highlighted in the main body of the manuscript.

Introduction:

Paragraph 2:

a) In the second paragraph, second line: does not this sentence “Persons with undiagnosed HIV or diagnosed but receiving medical care” should not be looked like: “Persons with undiagnosed HIV or diagnosed but NOT receiving medical care.”?

b) In the same sentence, this sentence “account for most new HIV transmissions” should be supported by some quantitative data or statistics.

Paragraph3:

In this sentence, “with the number of AIDS cases,” did the author mean “HIV and AIDS cases” or only AIDS cases? Make sure that these two in the current era (and even the past) are (were) different. If they rewrite all HIV cases, this may include those individuals in AIDS phase, but the other way round does not include HIV cases. Caution with regard to the use of AIDS and HIV terms should be made throughout the text.

Paragraph 7 (line 98-99): “and eligibility criteria reflect these criteria” is not clear; reflect which criteria?

I believe the Introduction should have another paragraph, before the Materials and Methods, to address the main objectives of this research and highlight the gaps for the current sampling strategy. From the rest of the introduction section, we found that there were two main designs for sampling, one from 2005 to 2014, and the other from 2015 to the present. But, what has been as limitations or has not been considered very well in these two designs, especially the present one, should be highlighted in the last paragraph when objectives are also listed.

Material and Method section:

Is there any possibility to better demonstrate Table 1 with the real values or statistics obtained from this research on how a final weight was constructed? Or, the authors may consider showing these steps in an MS Excel file how to construct such weights.

I believe that the authors should explain the reason(s) for each weight in a clear way. For example, “design weight” is a way to deal with sampling error (i.e., happens when the selected sample does not fully and accurately reflect sampling frame); “nonresponse weight” is a way to deal with nonresponse error which id defined as … .

Design weight paragraph (line 114):

The authors stated that the first component of the weight (design weight) was calculated as the reciprocal of the probability of selection (meaning the number of individuals in the frame population represented by each sample). While this is true, the authors should pay attention to the fact that this is true for the time when weights are incorporated based on a simple random sampling approach (i.e., equal inclusion probability for each individual in the sample frame). However, this is not true when other sampling strategies such a stratified random sampling or clustered random sampling might be necessary. In HIV research, and even in the context of this research, clustering sampling (where health facilities might be considered as clusters) is more common. How would different sampling strategies be accounted in this weighting process?

The definition of nonresponse adjustment/weigh (i.e., the reciprocal of the probability of responding) without considering the mechanism of missingness is incomplete. The probability that one (i) responds to the survey given that they were sample (Pr|s) is different when we assume missing completely at random, or missing at random (I think this is what the authors selected as their mechanism, but there are no clear justifications in the manuscript), or not missing at random. Such a mechanism should be explained clearly in this manuscript.

Nonresponse adjustment paragraph:

Comment on the logistic regression model: a) absolute log odds ratio (|log(OR)|) probably means the absolute value of beta estimate (log odds ratio) obtained from logistic regression. This should be clearer; b) in this sentence, “to choose the most significant predictors”, I believe “significant predictors” in the context of statistics and epidemiology refers to statistically significant, while the authors used beta coefficients to choose the “strongest predictors” as these beta coefficients were ranked. This should be clarified; c) to find the strongest predictors (if this is the case for what the authors did in this step), using the unstandardized format of the variable cannot identify these strong predictors. The standardized version of the included variables should be used to identify such strong predictors. This is another consideration in this step.

To enrich the results and methods section of this piece, I would strongly recommend the authors to provide their statistical models for both multivariable logistic regression and propensity weighting approach. These can be provided in both the formula and their software codes (syntaxes). Consider to explain and provide them in an APPENDIX.

I am wondering if how the authors incorporated the score obtained from propensity score approach in the weighting process? I mean, whether they just included the continuous measure of the score in the model, or they used a categorical format of the score, or whether they inverse-probability-censoring weighting (IPCW) approach? If IPCW was used, was it a unstabilized (i.e., 1/Pr(r|covariates)) or a stabilized (Pr(r)/Pr(r/covariates); where r refers to respondents). The stabilized approach has been commonly recommended to be used in case this is the approach for weighting process. The authors should clarify this process. In case they only used propensity score, they should also explain why they did not use stabilized IPCW.

Using a propensity based approach requires several key assumptions. How did the authors check and satisfy the assumption? For example, one key assumption of propensity approach is “Positivity” or “common support”, which is for each value of X (which is a vector of covariates), there is a positive probability of being both exposed and unexposed (in this manuscript, both responded and non-responded): 0 < Pr(r = 1|covariates) < 1. Another assumption is “conditional independence”; how the authors checked this assumption? “Correct model specification” is another assumption. “Unmeasured confounders” is another assumption.: The model is corrected, given unmeasured confounders. Therefore, a detailed explanation of these assumptions is definitely required.

Noncoverage adjustment paragraph:

a) How the noncoverage adjustment/weight were calculated? Was it based on the % of one stratum in the “target population” divided by % of the same stratum in the sample, or the % of one stratum in the “frame population” divided by % of the same stratum in the sample? While the total number of target population (both diagnosed and undiagnosed individuals living with HIV) is unknown (regardless the fact that they can be estimated), I believe this should be clarified in this section that the frame sample was used to calculate this weight;

b) There is a model-based approach (called Raking) using the totals of HIV diagnosed population (assuming that this is the sample frame) to create post-stratification weights so that the marginal values of a categorical variable add up to the totals. Given the efficiency of this approach in including many strata in the process of post-stratification adjustment, I am wondering why this approach had not been incorporated into the weighting process to better improve the novelty of the approach.

Line 192: stratified PPS design should be spelled out as PPS (probability proportional to size).

Variance estimation:

How the author defined non-certainty PSUs? As well, what about certainty PSUs? These two types of PSUs require more explanations.

Table 1 is not well-linked with the text in the methods section. For example, the formula provided for the nonresponse adjustment stage in Table 1 should be clearly and accurately explained in the text. what are those elements in the numerator and what are those in the denominator? Consider explaining all these elements in the text.

Again, I strongly recommend the authors to provide a detailed process of the sampling when explaining each section of the methods and subsequently, the results section. Considering to have an MS Excel file (as an APPENDIX) explaining each step in a practical way is highly recommended. This can help the authors absorb more audiences who can evaluate their great research work, and probably improve it in the future. This can help the manuscript be an applied paper such that researchers in other fields (non-HIV fields) can also consider this approach. In each step they provide information, a comparison of the approaches (the one the authors recommended and the previous approach of sampling) is also recommended.

Results:

The author should provide an N for their sample frame; 9,700 persons were sampled from what N?

In Fig 2: both n and N are better to be reported in each step. For example, 9179/9700 (94.6%) were eligible; 5113/9179 (55.7%) of eligibles were contacted; 3654/9179 (39.8%) of eligibles responded. In addition, is there any possibility of adding ranges to each % in each step in this fig 2? If yes, I would also recommend to add them.

Results paragraph 2:

This sentence, “Participation was 71.5% nationally” is not in line with what can be seen from Fig 2. This is true that the conditioning populations are different, but I would recommend having the same numbers in both Table 2 and text. 71.5% was among those contacted, but this is misleading (an appreciably high %). This should be also reported by the total eligibles (which is 39.8%, with range ##, ##).

Results Nonresponse adjustment:

strongest predictors and the beta coefficients were identified and obtained using an unstandardized format of the variable (right?). If yes, this approach cannot identify the strong predictors as each predictor has its own specific unit (not consistent units). To identify the strong predictors, standardized approach should be used. This standardized approach is not typically recommended in health sciences, but for the case of this research and to identify factors that are strongly associated with response to the survey (not based on P-value, rather based on the magnitude of the estimate), standardized coefficients should be used.

“presumed HIV care status” should be better explained and defined as this is the single factor remained in the model to be a strong predictor of response. How “frequency of HIV lab results” was defined? Was it a count variable: 0 result, 1 result, 2 results, 3 results, and so one? What did 0 results indicate and what other results indicated.

Which “model fit statistics” were used to compare both weighting class and propensity model?

The authors chose the cell (class) weighting method “because, having less variance, it performed better than the propensity method.” This means nothing unless there are statistics and some findings to support such a claim. Without these statistics, we cannot evaluate or even trust why class weighting performed better.

The authors believe that class weighting was less variant: a) first less variant requires some quantitative supports, b) how much less was considered to be performing better? This is why I believe details should be reported as the journal does not have any limitation on the number of pages or words.

The whole results and methods sections are bare, meaning that the authors, unfortunately, did not provide details of their approaches to cover their methods, and required statistics and results to support their claim. A detailed explanation and description of the method section as well as detailed provision of results are required.

Results last paragraph:

“produced similar results” requires detailed statistics. What does “similar” mean in this context?

The authors believe that “Because implementing a two-stage adjustment was generally not feasible locally, we opted for a single-stage adjustment nationally”: is this also a new or innovative part of the results section? I am wondering which one of these two (single-stage or two stages) was performed in the previous cycles? As the two-stage was not feasible locally in this new version of the weighting, I think it had also not been feasible previously, right? So, this is not really a surprising result of this manuscript? Please provide explanations on this too.

Discussion and conclusion:

The authors should pay attention to the point that performing multiple weighting steps in surveys is to improve the REPRESENTATIVENESS of the selected sample. Given this, do the authors believe that a two-stage approach (when both noncontact and nonresponse were considered in computing weights) could not provide a more accurate estimate representative to the frame sample than a single-stage? I am wondering who much their findings and suggestions are based on the type of data they use (i.e., a data-driven approach) than the evidence supported by statistical theories (i.e., a theory-supported approach) that having multiple or sequential stages in weighting can better provide representative estimates? This should be also discussed in the discussion section.

6. PLOS authors have the option to publish the peer review history of their article (what does this mean?). If published, this will include your full peer review and any attached files.

Reviewer #1: Yes: Mostafa Shokoohi

---

## [Author Response · Author response to Decision Letter 0]

25 Feb 2020

We thank the review for their comments. We have edited the manuscript and feel it has been strengthened as a result. Our point-by-point response is below:

Reviewer: This research aimed to describe methods used for computing survey weights, the approach chosen, and the benefits of using a dynamic surveillance registry as a sampling frame for adults living with HIV from the National HIV Surveillance System in the United States. While the authors have reported an excellent research topic, I believe that the following comments can be also helpful to consider in their revised version. What I believe this manuscript lacks are bare explanations of the methods used in the Methods section and bare presentation of the Results section. The authors should provide a detailed description and presentation of their findings. 

Title: 

I believe the title should show the area of this research. This is why I think we should have an “HIV” word somewhere like, “… in order to include persons not receiving HIV medical care”. As well, the title can be shorter. For example, “in order to include persons not receiving” can be written as “to also include persons not receiving”. The word “also” (or any other alternative) in this suggested title can reflect that the system had not included such individuals, but these should be also included along with those who receive HIV care. 

Response: We edited the title to incorporate these suggestions, although we feel that the phrasing “changes … to include” conveys that such persons were previously excluded.

We are unsure how to address some of the overarching comments, which are at times contradictory. For example, the reviewer suggests that the manuscript lacks “bare explanations” and “bare presentation of results,” but also advises us to provided “a detailed descriptions and presentation.” In our expositon we have attempted to provide a clear explanation of the process we followed without excessive detail, since we employed methods that are not in themselves novel and have been written about extensively (and in sufficient detail to inform those unfamiliar with the methods) by others. We feel that we have not omitted any important step or decision point.

Reviewer: Abstract: 

When the authors claim that “weighting class adjustments and a single-stage nonresponse adjustment performed best,” they have to be more specific in what regard these methods turn to be the “best.” Or, “strongly associated with” does not reflect anything unless we see some outputs or even some key findings. Please avoid using such vague sentences and provide more evidence for these results. 

Response: We justify these claims in the Results section and for brevity summarize them in the abstract, where we lack room to provide specifics of the comparison. We have edited the first sentence to tone down its comparative aspect by rephrasing it as “After assessing these methods, we chose as our preferred procedure weighting class adjustments and a single-stage nonresponse adjustment.” We then modified the following sentence to clarify that “Classes were constructed using variables associated with respondents’ characteristics and important survey outcomes, chief among them laboratory results available from surveillance that served as a proxy for medical care.”

Reviewer: The abstract does not seem to be a good place talking about how the results of this HIV case study might also have implications for “other disease registries”. This can be highlighted in the main body of the manuscript. 

Response: We prefer to keep this sentence in the abstract because we feel that it is important to highlight the utility of the manuscript findings for other research, programs, and systems. It also emphasizes the value of supplemental surveillance (in which a sample survey leverages data from a disease registry) in public health practice.

Reviewer: Introduction: 

Paragraph 2: 

a) In the second paragraph, second line: does not this sentence “Persons with undiagnosed HIV or diagnosed but receiving medical care” should not be looked like: “Persons with undiagnosed HIV or diagnosed but NOT receiving medical care.”? 

Response: This error has been corrected.

Reviewer: b) In the same sentence, this sentence “account for most new HIV transmissions” should be supported by some quantitative data or statistics.

Response: We added the specific estimates from the cited article. 

Reviewer: Paragraph3: 

In this sentence, “with the number of AIDS cases,” did the author mean “HIV and AIDS cases” or only AIDS cases? Make sure that these two in the current era (and even the past) are (were) different. If they rewrite all HIV cases, this may include those individuals in AIDS phase, but the other way round does not include HIV cases. Caution with regard to the use of AIDS and HIV terms should be made throughout the text. 

Response: The sampling was based on AIDS cases and did not include persons with HIV infection who had not progressed to AIDS. This was because HIV non-AIDS diagnoses were not reportable in all U.S. jurisdictions at the time of sampling, early in the project. 

Reviewer: Paragraph 7 (line 98-99): “and eligibility criteria reflect these criteria” is not clear; reflect which criteria? 

Response: The sentence has been edited to specify the criteria, i.e., alive, diagnosed with HIV, aged 18 years or older, residing in the US. 

Reviewer: I believe the Introduction should have another paragraph, before the Materials and Methods, to address the main objectives of this research and highlight the gaps for the current sampling strategy. From the rest of the introduction section, we found that there were two main designs for sampling, one from 2005 to 2014, and the other from 2015 to the present. But, what has been as limitations or has not been considered very well in these two designs, especially the present one, should be highlighted in the last paragraph when objectives are also listed. 

Response: We have added a paragraph to the end of the introduction. 

Reviewer: Material and Method section: 

Is there any possibility to better demonstrate Table 1 with the real values or statistics obtained from this research on how a final weight was constructed? Or, the authors may consider showing these steps in an MS Excel file how to construct such weights. 

I believe that the authors should explain the reason(s) for each weight in a clear way. For example, “design weight” is a way to deal with sampling error (i.e., happens when the selected sample does not fully and accurately reflect sampling frame); “nonresponse weight” is a way to deal with nonresponse error which id defined as … . 

Response: We have added sentences, or added phrases to existing sentences, to say something about the purpose of each component of the weights described in this section. 

The design weights account for unequal sampling fractions when a sample is something other than a simple random sample. Their use is warranted when the sample is deliberately selected in a way that causes it to depart from a proportionate representation of the frame (e.g. when sampling fractions vary by state in recognition of different population sizes. Our exposition lays out the adjustment stages we employed sequentially in constructing weights and the purpose of each.

Reviewer: Design weight paragraph (line 114): 

The authors stated that the first component of the weight (design weight) was calculated as the reciprocal of the probability of selection (meaning the number of individuals in the frame population represented by each sample). While this is true, the authors should pay attention to the fact that this is true for the time when weights are incorporated based on a simple random sampling approach (i.e., equal inclusion probability for each individual in the sample frame). However, this is not true when other sampling strategies such a stratified random sampling or clustered random sampling might be necessary. In HIV research, and even in the context of this research, clustering sampling (where health facilities might be considered as clusters) is more common. How would different sampling strategies be accounted in this weighting process? 

Response: The design weight is by definition the reciprocal of the probability of selection even when the sample is stratified or clustered. In those cases, the selection probabilities vary by stratum or cluster, and the design weights also vary by stratum or cluster. This is also the case in multi-stage sampling, which is relevant to our situation because the poulations of MMP sites vary greatly, while the sample sizes by project vary much less.

Cluster sampling by facilities is indeed common in HIV research, and in fact was part of the previous MMP sample design, before the transition to using surveillance registries as the frame. That design, the rationale for it, and the weighting methods we employed in conformance with the design, are described in other publications. We do not see the need to describe how we would account in our weighting process for a sampling strategy we did not employ.

Reviewer: The definition of nonresponse adjustment/weigh (i.e., the reciprocal of the probability of responding) without considering the mechanism of missingness is incomplete. The probability that one (i) responds to the survey given that they were sample (Pr|s) is different when we assume missing completely at random, or missing at random (I think this is what the authors selected as their mechanism, but there are no clear justifications in the manuscript), or not missing at random. Such a mechanism should be explained clearly in this manuscript. 

Response: The Missing At Random (MAR) or Missing Completely At Random (MCAR) distinction is a foundational issue that even most technical discussions of implementing weighting methods ignore, and it does not seem appropriate to discuss a largely theoretical matter in describing our applied work. The use of nonresponse adjustments based on comparing the characteristics of respondents and nonrespondents is an implicit concession that (unit) response is not MCAR. By default, both the adjustment methods we considered were MAR. 

Reviewer: Nonresponse adjustment paragraph: 

Comment on the logistic regression model: a) absolute log odds ratio (|log(OR)|) probably means the absolute value of beta estimate (log odds ratio) obtained from logistic regression. This should be clearer; b) in this sentence, “to choose the most significant predictors”, I believe “significant predictors” in the context of statistics and epidemiology refers to statistically significant, while the authors used beta coefficients to choose the “strongest predictors” as these beta coefficients were ranked. This should be clarified; c) to find the strongest predictors (if this is the case for what the authors did in this step), using the unstandardized format of the variable cannot identify these strong predictors. The standardized version of the included variables should be used to identify such strong predictors. This is another consideration in this step. 

Response: We agree that it is clearer to say that we chose “the strongest predictors” rather than “the most predictors,” and is also more consistent with the following sentence about ranked |log(OR)|s, and have edited the phrase. Our intent was not to establish epidmiologic significance but instead to choose among variables for a model. The issue of the standardized format of variables was not a concern for us because the predictors of response that we chose were overwhelmingly categorical, and in no case did we need to choose among continuous predictors

Reviewer: To enrich the results and methods section of this piece, I would strongly recommend the authors to provide their statistical models for both multivariable logistic regression and propensity weighting approach. These can be provided in both the formula and their software codes (syntaxes). Consider to explain and provide them in an APPENDIX. 

Response: We assume that the reader has some familiarity with logistic regression, which underlies these methods, and do not feel that providing formulas would be helpful. (We did, for comparison, feel that providing formulas for the different weights applied was essential and helps the reader understand the components of the weights as well as the sequence and levels of analysis in the process). As for the software employed, the SAS and SUDAAN procedures are themselves straightforward applications, but because they are embedded in complex macros, they employ many macro variables and conditionally-executed code that would be difficult for someone new to the process to decipher without expending considerable effort.

Reviewer: I am wondering if how the authors incorporated the score obtained from propensity score approach in the weighting process? I mean, whether they just included the continuous measure of the score in the model, or they used a categorical format of the score, or whether they inverse-probability-censoring weighting (IPCW) approach? If IPCW was used, was it a unstabilized (i.e., 1/Pr(r|covariates)) or a stabilized (Pr(r)/Pr(r/covariates); where r refers to respondents). The stabilized approach has been commonly recommended to be used in case this is the approach for weighting process. The authors should clarify this process. In case they only used propensity score, they should also explain why they did not use stabilized IPCW. 

Response: We grouped the propensities into categories (usually quintiles) to reduce their variability. This is mentioned in the “adjustment method” paragraph and referred to again in the “nonresponse adjustment” paragraph of the Results section.

We did not consider the IPCW approach, and are familiar with its use only in the context of survival analysis, not as a part of propensity weighting for survey nonresponse adjustment. Both the weighting class and propensity weighting methods assume a dichotomous outcome rather than different mathematical models that incorporates a time-to-event covariate. Nor do we have censored data (unless one were to treat the end of the field period as the censoring event, in which case only those who were contacted and refused were true non-censored nonrespondents, but we are unaware of such an approach being applied in nonresponse adjustment for survey weighting).

Reviewer: Using a propensity based approach requires several key assumptions. How did the authors check and satisfy the assumption? For example, one key assumption of propensity approach is “Positivity” or “common support”, which is for each value of X (which is a vector of covariates), there is a positive probability of being both exposed and unexposed (in this manuscript, both responded and non-responded): 0 <Pr(r = 1|covariates) < 1. Another assumption is “conditional independence”; how the authors checked this assumption? “Correct model specification” is another assumption. “Unmeasured confounders” is another assumption.: The model is corrected, given unmeasured confounders. Therefore, a detailed explanation of these assumptions is definitely required. 

Response: The reviewer brings up basic mathematical concepts underlying logistic regression models in general. The assumptions of conditional independence and unmeasured confounders are not required for the weighting class method, which equates to a cell-means model in regression. The classes are constructed in such a way that the number of respondents and nonrespondents is not small, so satisifying the probability of being either exposed or unexposed would be automatic. Ours was a routine application of methods that are applied frequently within survey practice, and nothing unique about our situation would preclude the use of either technique. Because of that, we see no reason to explain the underlying theory.

Reviewer: Noncoverage adjustment paragraph: 

a) How the noncoverage adjustment/weight were calculated? Was it based on the % of one stratum in the “target population” divided by % of the same stratum in the sample, or the % of one stratum in the “frame population” divided by % of the same stratum in the sample? While the total number of target population (both diagnosed and undiagnosed individuals living with HIV) is unknown (regardless the fact that they can be estimated), I believe this should be clarified in this section that the frame sample was used to calculate this weight; 

Response: This is explained in the second paraphgraph, when we refer to the updated counts and post-stratifying so that totals match. The corresponding formula in the table shows that this adjustment is the ratio of totals.

Reviewer: b) There is a model-based approach (called Raking) using the totals of HIV diagnosed population (assuming that this is the sample frame) to create post-stratification weights so that the marginal values of a categorical variable add up to the totals. Given the efficiency of this approach in including many strata in the process of post-stratification adjustment, I am wondering why this approach had not been incorporated into the weighting process to better improve the novelty of the approach. 

Response: Among calibration methods, survey often employ raking when external control totals are used as to quantify the target population, particularly when these external controls are derived from independent sources. We used only auxiliary variables from the surveillance system, and thus were able to calculate cells in the cross-tabulation of post-stratification variables directly rather than having to estimate them from marginal proportions, as is the case with raking, which would risk increasing bias for some demographic subgroups. Rather than employing novel approaches for their own sake, we chose throughout our work to apply standard techniques consistent with past practice. Choosing unobjectionable, defensible methods while maintaining as much methodological continuity as was reasonable motivated us, given the many other changes to the sample design.

Reviewer: Line 192: stratified PPS design should be spelled out as PPS (probability proportional to size). 

Response: This has been edited.

Reviewer: Variance estimation: 

How the author defined non-certainty PSUs? As well, what about certainty PSUs? These two types of PSUs require more explanations. 

Response: Selection of PSUs was described previously, in the first paragraph of the section titled “2005 – 2014 population, frame, and sample design.” We have added a phrase there to clarify that some PSUs were sampled with certainty.

Reviewer: Table 1 is not well-linked with the text in the methods section. For example, the formula provided for the nonresponse adjustment stage in Table 1 should be clearly and accurately explained in the text. what are those elements in the numerator and what are those in the denominator? Consider explaining all these elements in the text. 

Response: The order of the narrative in this part of the text parallels the elements of the table as well as the stages of adjustment that are applied sequentially. Each component we describe is itself a standard application of the form of adjustment applied, and the paragraph describing that component summarizes its purpose. The table uses notation to clarify which group is subject to the adjustments via subscripts and set inclusion notation. For these reasons, we feel that adding to the narrative to describe every element of every formula would decrease readability for the reader without making it clearer.

Reviewer: Again, I strongly recommend the authors to provide a detailed process of the sampling when explaining each section of the methods and subsequently, the results section. Considering to have an MS Excel file (as an APPENDIX) explaining each step in a practical way is highly recommended. This can help the authors absorb more audiences who can evaluate their great research work, and probably improve it in the future. This can help the manuscript be an applied paper such that researchers in other fields (non-HIV fields) can also consider this approach. In each step they provide information, a comparison of the approaches (the one the authors recommended and the previous approach of sampling) is also recommended. 

Response: Sampling in MMP is unusually simple for a national survey. As described in the section titled “2015 – present population, frame, and sample design,” in the paragraph following Figure 1, CDC staff draws simple random samples from each of the 23 separate frame files. There is no further detail to provide.

We appreciate the reviewer’s kind remarks complimenting our work and concern for popularizing it. However, researchers wishing to apply our work to their own situations, which is indeed our hope, will need considerably more background than a manuscript such as this, and we have provided the references we think would be most helpful for acquiring such background (including other publications describing MMP sampling and weighting methods, allowing us to focus in the present manuscript on what is new since the recent changes to the MMP population and frame). We are unsure how providing this information in spreadsheet form would be helpful; much of that, it seems to us, would repeat the table with weight components and accompanying narrative section.

Reviewer: Results: 

The author should provide an N for their sample frame; 9,700 persons were sampled from what N?

Response: We have added this information. 

Reviewer: In Fig 2: both n and N are better to be reported in each step. For example, 9179/9700 (94.6%) were eligible; 5113/9179 (55.7%) of eligibles were contacted; 3654/9179 (39.8%) of eligibles responded. In addition, is there any possibility of adding ranges to each % in each step in this fig 2? If yes, I would also recommend to add them. 

Response: We feel that adding numerators and denominators for each proportion mentioned in the figure is not needed because the denominator is easily inferred from the preceding step, and the reader can easily reproduce the proportions cited. Adding ranges across states would make it harder to follow. 

Reviewer: Results paragraph 2: 

This sentence, “Participation was 71.5% nationally” is not in line with what can be seen from Fig 2. This is true that the conditioning populations are different, but I would recommend having the same numbers in both Table 2 and text. 71.5% was among those contacted, but this is misleading (an appreciably high %). This should be also reported by the total eligibles (which is 39.8%, with range ##, ##). 

Response: In the sentence referred to, we define participation as response among those contacted. This is ratio of the 3,654 eligibles responded and the 5,113 eligibles contacted (3654/5113=0.7146=71.5%. We have edited the sentence to provide these two counts. We changed the term “participation” to “cooperation,” so that the rate we present is consistent with Cooperation Rate 2 (COOP2) as defined in AAPOR’s standard definitions. 

Reviewer: Results Nonresponse adjustment: 

strongest predictors and the beta coefficients were identified and obtained using an unstandardized format of the variable (right?). If yes, this approach cannot identify the strong predictors as each predictor has its own specific unit (not consistent units). To identify the strong predictors, standardized approach should be used. This standardized approach is not typically recommended in health sciences, but for the case of this research and to identify factors that are strongly associated with response to the survey (not based on P-value, rather based on the magnitude of the estimate), standardized coefficients should be used. 

Response: To our knowledge, standardized coefficients are not commonly used in this particular situation of choosing the strongest predictors of survey response among categorical predictors. Our models overwhelmingly led to the use of categorical variables, so standardizing coefficients was not a concern. In no case were we in the situation of choosing between even two continuous variables, in which case standardizing their scales might have informed the comparison.

Reviewer: “presumed HIV care status” should be better explained and defined as this is the single factor remained in the model to be a strong predictor of response. How “frequency of HIV lab results” was defined? Was it a count variable: 0 result, 1 result, 2 results, 3 results, and so one? What did 0 results indicate and what other results indicated. 

Response: We have edited the text to clarify that (while we did indeed have a count variable) we used a three-level indicator for any care; this was measured by the presence or absence of HIV lab results in the person’s surveillance records over the past 12 months: 2+ HIV labs in the past 12 months 90 days or more apart, at least 1 HIV lab in the past 12 months, and 0 HIV labs in the past 12 months.

Reviewer: Which “model fit statistics” were used to compare both weighting class and propensity model? 

The authors chose the cell (class) weighting method “because, having less variance, it performed better than the propensity method.” This means nothing unless there are statistics and some findings to support such a claim. Without these statistics, we cannot evaluate or even trust why class weighting performed better. 

The authors believe that class weighting was less variant: a) first less variant requires some quantitative supports, b) how much less was considered to be performing better? This is why I believe details should be reported as the journal does not have any limitation on the number of pages or words. 

The whole results and methods sections are bare, meaning that the authors, unfortunately, did not provide details of their approaches to cover their methods, and required statistics and results to support their claim. A detailed explanation and description of the method section as well as detailed provision of results are required. 

Response: 

We are including, as an appendix of the revised manuscript, a table of AUC statistics. These aided us in evaluting the methods compared and quantify the sort of differences the reviewer mentions. We had no formal threshold for “how much less was considered better” because this was only one of several factors we considered when comparing the two methods. 

As mentioned previously, because ours is a standard application of methods that have been extensively used and written about, we feel that we have provided sufficient detail (and references for those who may require more background).

Reviewer: Results last paragraph: 

“produced similar results” requires detailed statistics. What does “similar” mean in this context? 

The authors believe that “Because implementing a two-stage adjustment was generally not feasible locally, we opted for a single-stage adjustment nationally”: is this also a new or innovative part of the results section? I am wondering which one of these two (single-stage or two stages) was performed in the previous cycles? As the two-stage was not feasible locally in this new version of the weighting, I think it had also not been feasible previously, right? So, this is not really a surprising result of this manuscript? Please provide explanations on this too. 

Response: We have provided additional information in an appendix relevant to this judgment (which was, admittedly, based on general impressions and not purely quantative). Also, the remainder of this paragraph describes other considerations that were ultimately greater concerns.

The multi-stage approach was not performed (nor considered) in the previous cycles, and thus there is no historical comparison to make. Because of the clustered design, it would have been difficult, and probably infeasible, before.

Reviewer: Discussion and conclusion: 

The authors should pay attention to the point that performing multiple weighting steps in surveys is to improve the REPRESENTATIVENESS of the selected sample. Given this, do the authors believe that a two-stage approach (when both noncontact and nonresponse were considered in computing weights) could not provide a more accurate estimate representative to the frame sample than a single-stage? I am wondering who much their findings and suggestions are based on the type of data they use (i.e., a data-driven approach) than the evidence supported by statistical theories (i.e., a theory-supported approach) that having multiple or sequential stages in weighting can better provide representative estimates? This should be also discussed in the discussion section.

Response: We have added sentences to the relevant paragraph in the discussion section to address this insightful point. In most of this kind of work, there is often a tension between theoretical considerations and data-driven approaches, and that was certainly the case here.

---

## [Decision Letter · Decision Letter 1]

26 Oct 2020

PONE-D-19-17946R1

Changes to the sample design and weighting methods of a public health surveillance system to also include persons not receiving HIV medical care

PLOS ONE

Dear Dr. Linda Beer,

Thank you for submitting your manuscript to PLOS ONE. After careful consideration, we feel that it has merit but does not fully meet PLOS ONE’s publication criteria as it currently stands. Therefore, we invite you to submit a revised version of the manuscript that addresses the points raised during the review process.

We look forward to receiving your revised manuscript.

Kind regards,

Mohammad Asghari Jafarabadi

Academic Editor

PLOS ONE

Reviewers' comments:

Reviewer's Responses to Questions

**Comments to the Author**

1. If the authors have adequately addressed your comments raised in a previous round of review and you feel that this manuscript is now acceptable for publication, you may indicate that here to bypass the “Comments to the Author” section, enter your conflict of interest statement in the “Confidential to Editor” section, and submit your "Accept" recommendation.

Reviewer #2: (No Response)

2. Is the manuscript technically sound, and do the data support the conclusions?

Reviewer #2: Yes

3. Has the statistical analysis been performed appropriately and rigorously? 

Reviewer #2: No

4. Have the authors made all data underlying the findings in their manuscript fully available?

Reviewer #2: No

5. Is the manuscript presented in an intelligible fashion and written in standard English?

Reviewer #2: Yes

6. Review Comments to the Author

Reviewer #2: Please find the enclosed file for the comments. Thx

7. PLOS authors have the option to publish the peer review history of their article (what does this mean?). If published, this will include your full peer review and any attached files.

Reviewer #2: No

---

## [Author Response · Author response to Decision Letter 1]

10 Nov 2020

Thank you for the opportunity to revise this manuscript. Below is a point-by-point response to the reviewer’s comments.

Comments to the authors:

This study is about calculating survey weights, the approach chosen, and the benefits of using a dynamic surveillance registry as a sampling frame, an exciting study. However, many shortcomings should be addressed/implemented.

Here are my detailed comments with special attention in methodological parts:

ABSTRACT 

• State the design of the study clearly.

• MMP is described in the Objectives paragraph of the abstract as a surveillance system. Elsewhere in the abstract, we refer to its sample frame and design. The focus of our paper is on weighting methods, so we devote most of the text of the abstract to describing components of the weights and our comparison procedure, with references to study decide provided as needed in the text to motivate the discussion of weighing.

• State the source of subjects stated.

• The Methods paragraph of the abstract, first sentence, identifies the HIV case surveillance registry as the source of study subject.

• Present the results using suitable statistical measures with CI's.

• Because this was a comparison of different weighting methods, results are not well summarized by particular statistical measures and their associated CIs. Unlike manuscripts with a single outcome (or even a small number of key outcomes), this manuscript is different in that we summarize methods for calculating weights use in all analyses of MMP data, whichever outcome might be the focus.

• State the implications of key findings as conclusions based on the results. The last sentence could not be concluded thru the obtained results.

• The last sentence is a recapitulation of the definition of supplemental surveillance. We have added text after that to clarify that the more-detailed information (the supplemental information), which would be cost-prohibitive to obtain from all subjects, can instead be collected through a sample survey of a portion of subjects, which is the method described in the paper.

INTRODUCTION

• Explain the gap of knowledge and necessity of the study precisely.

• We have added two sentences to the introduction clarifying that this extended discussion of weighting in MMP has not previously been available, and how this information can inform other studies that use sample survey methods.

METHODS

• Divide the methods section into determined subsections with defined subheadings according to the STROBE statement.

• Most elements of the STROBE rubric are not well suited to the kind of methodologic work that is the basis of this manuscript; they are, rather, more appropriate for reporting results of individual epidemiologic studies (such as particular instances of topic- or subpopulation-specific analyses of MMP topics). Instead, as appropriate, we have provided response rates that are defined by CASRO, which recommends their inclusion and citation in manuscripts. These fundamental concerns of survey operations are a more appropriate framework for reporting on methodological details such as these.

• State the design of the study and its key elements clearly.

• The Methods section describes these in great detail.

• State the setting of the study clearly.

• We describe both the national study population and the states participating, which themselves constitute populations, as well as the roles of the CDC and the state and local health departments involved.

• State the relevant dates of study, including periods of recruitment and data collection.

• We provide the study reference period, the data collection cycle, and other timing consideration for this ongoing surveillance project.

• State the source of subjects.

• We describe the study population and the frame, derived from surveillance records, used to represent it.

• State the number of subjects.

• We provide both frame and sample sizes in the Results section.

• Explain the sampling design and procedures in detail.

• The sampling design is explained in considerable detail, as is necessary to motivate the discussion of the design weight stage of weighting and variance estimation procedure and how these are consistent with the sample design.

• State the study variables (including outcomes, predictors, and potential confounders).

• We have described the relevant outcomes (noncontact, nonresponse) at each stage of the weighting process and the predictors considered, as well as the rationale for considering a variable as a predictor. Confounding is not really a concern in the nonresponse modeling process for the weighting approaches we considered.

• State the name, version, and address of the software.

• Although most calculations were carried out using SAS, such specifics seem not especially relevant, since we are describing procedures that could be implemented in almost any statistical package.

• Describe the statistical analytical methods taking account of the sampling strategy. 

• Components of the analysis weights follow the sampling design at every stage, and these correspondences are thoroughly explained in the exposition.

• Describe the sensitivity analyses clearly. 

• As with confounding, sensitivity analysis is not applicable for the weighting methods considered.

• Mention the ethics code. 

• MMP is considered public health practice and is exempt from human subjects review, but undergoes IRB review routinely where local regulation require it. This is described on pages 13-14.

RESULTS

• Present the characteristics of study participants (e.g., demographic, clinical, and social) and information on exposures and potential confounders. 

• In this methodological work, we have described how information on sampled subjects is used in making population-level estimates. Characterizing only study participants (respondents) would be insufficient, since weighting methods (which we describe) compare the characteristics of both respondents and nonrespondents to compensate for nonresponse bias. The scope of topics included in MMP is broad, and exposures and potential confounders appropriate to consider would depend on the particular outcome under study, which is not the purpose of this manuscript. 

• Present the table of demographic characteristics.

• Because the purpose of this manuscript is to describe weighting methods, rather than characterizing the sample or the study population, we describe which characteristics are considered as weighting factors, along with the rationale (the conditions they must satisfy to be included in the weighting). Reference 2 in the manuscript describes the demographic characteristics of MMP respondents. 

• Summarize and describe the outcome measures of the study with suitable statistical measures and their CI.

• Because the focus of this manuscript is weighting methods, one might think of contact and response as outcomes, and we have described how these are incorporated into the weighting process. Otherwise, this point seems not entirely relevant to the topic of weight construction.

DISCUSSION

• Discuss the generalizability (external validity, applicability) of the trial findings that.

• This comment is unclear – was the sentence truncated? This manuscript describes weighting methods and is not a reporting of trial findings. We do note in the “Public Health Implications” section that sample surveys using MMP methods could be feasible for supplemental surveillance in other disease registries and population-monitoring systems whose timeliness and completeness are established. 

• Recommend further studies in the future. 

• No further refinements to the MMP sample design or to the weighting procedures currently employed are currently being considered, although project staff continue to monitor response rates during data collection, as well as predictors of response during the annual weighting process, in case any refinements may become necessary.

---

## [Decision Letter · Decision Letter 2]

16 Nov 2020

PONE-D-19-17946R2

Changes to the sample design and weighting methods of a public health surveillance system to also include persons not receiving HIV medical care

PLOS ONE

Dear Dr. Beer,

Thank you for submitting your manuscript to PLOS ONE. After careful consideration, we feel that it has merit but does not fully meet PLOS ONE’s publication criteria as it currently stands. Therefore, we invite you to submit a revised version of the manuscript that addresses the points raised during the review process.

We look forward to receiving your revised manuscript.

Kind regards,

Mohammad Asghari Jafarabadi

Academic Editor

PLOS ONE

**Reviewers' comments:**

Reviewer's Responses to Questions

**Comments to the Author**

1. If the authors have adequately addressed your comments raised in a previous round of review and you feel that this manuscript is now acceptable for publication, you may indicate that here to bypass the “Comments to the Author” section, enter your conflict of interest statement in the “Confidential to Editor” section, and submit your "Accept" recommendation.

Reviewer #2: All comments have been addressed

2. Is the manuscript technically sound, and do the data support the conclusions?

Reviewer #2: Partly

3. Has the statistical analysis been performed appropriately and rigorously? 

Reviewer #2: Yes

4. Have the authors made all data underlying the findings in their manuscript fully available?

Reviewer #2: Yes

5. Is the manuscript presented in an intelligible fashion and written in standard English?

Reviewer #2: Yes

6. Review Comments to the Author

Reviewer #2: The comments have been well addressed, some minor points:

1- In the abstarct and methods section add the design of the shtudy as "methodological study".

2- A description of participnats' profile (although the authors are not agreed to add the demographic table), should be explained at the begiing of the results section.

7. PLOS authors have the option to publish the peer review history of their article (what does this mean?). If published, this will include your full peer review and any attached files.

Reviewer #2: No

---

## [Author Response · Author response to Decision Letter 2]

17 Nov 2020

Thank you for the review. Our responses to the reviewer’s comments are listed below.

Reviewer #2: The comments have been well addressed, some minor points:

1- In the abstarct and methods section add the design of the shtudy as "methodological study".

Response: This has been added to the abstract and methods section (lines 35-36 and 119 in tracked changes version of manuscript).

2- A description of participnats' profile (although the authors are not agreed to add the demographic table), should be explained at the begiing of the results section.

Response: A brief description of the participant characteristics and a reference to the document in which they are described in full has been added to the results section (lines 222-226 in tracked changes version of manuscript).

---

## [Decision Letter · Decision Letter 3]

20 Nov 2020

Changes to the sample design and weighting methods of a public health surveillance system to also include persons not receiving HIV medical care

PONE-D-19-17946R3

Dear Dr. Beer,

We’re pleased to inform you that your manuscript has been judged scientifically suitable for publication and will be formally accepted for publication once it meets all outstanding technical requirements.

Kind regards,

Mohammad Asghari Jafarabadi

Academic Editor

PLOS ONE

Reviewers' comments:

Reviewer's Responses to Questions

**Comments to the Author**

1. If the authors have adequately addressed your comments raised in a previous round of review and you feel that this manuscript is now acceptable for publication, you may indicate that here to bypass the “Comments to the Author” section, enter your conflict of interest statement in the “Confidential to Editor” section, and submit your "Accept" recommendation.

Reviewer #2: All comments have been addressed

2. Is the manuscript technically sound, and do the data support the conclusions?

Reviewer #2: Yes

3. Has the statistical analysis been performed appropriately and rigorously? 

Reviewer #2: Yes

4. Have the authors made all data underlying the findings in their manuscript fully available?

Reviewer #2: Yes

5. Is the manuscript presented in an intelligible fashion and written in standard English?

Reviewer #2: Yes

6. Review Comments to the Author

Reviewer #2: Thanks for revising the manuscript properly.

....................................................................

7. PLOS authors have the option to publish the peer review history of their article (what does this mean?). If published, this will include your full peer review and any attached files.

Reviewer #2: No

---

## [Editor Report · Acceptance letter]

23 Nov 2020

PONE-D-19-17946R3 

Changes to the sample design and weighting methods of a public health surveillance system to also include persons not receiving HIV medical care 

Dear Dr. Beer:

I'm pleased to inform you that your manuscript has been deemed suitable for publication in PLOS ONE. Congratulations! Your manuscript is now with our production department. 

Kind regards, 

on behalf of

Professor Mohammad Asghari Jafarabadi 

Academic Editor

PLOS ONE